# Online Learning with Costly Features and Labels

**Navid Zolghadr**
Department of Computing Science
University of Alberta
zolghadr@ualberta.ca

**Gábor Bartók**
Department of Computer Science
ETH Zürich
bartok@inf.ethz.ch

**Russell Greiner**    **András György**    **Csaba Szepesvári**
Department of Computing Science, University of Alberta
{rgreiner,gyorgy,szepesva}@ualberta.ca

## Abstract

This paper introduces the *online probing* problem: In each round, the learner is able to purchase the values of a subset of feature values. After the learner uses this information to come up with a prediction for the given round, he then has the option of paying to see the loss function that he is evaluated against. Either way, the learner pays for both the errors of his predictions and also whatever he chooses to observe, including the cost of observing the loss function for the given round and the cost of the observed features. We consider two variations of this problem, depending on whether the learner can observe the label for free or not. We provide algorithms and upper and lower bounds on the regret for both variants. We show that a positive cost for observing the label significantly increases the regret of the problem.

## 1  Introduction

In this paper, we study a variant of online learning, called *online probing*, which is motivated by practical problems where there is a cost to observing the features that may help one's predictions. Online probing is a class of online learning problems. Just like in standard online learning problems, the learner's goal is to produce a good predictor. In each time step $t$, the learner produces his prediction based on the values of some feature $x_t = (x_{t,1}, \ldots, x_{t,d})^\top \in \mathcal{X} \subset \mathbb{R}^d$.[1] However, unlike in the standard online learning settings, if the learner wants to use the value of feature $i$ to produce a prediction, he has to purchase the value at some fixed, *a priori* known cost, $c_i \geq 0$. Features whose value is not purchased in a given round remain unobserved by the learner. Once a prediction $\hat{y}_t \in \mathcal{Y}$ is produced, it is evaluated against a loss function $\ell_t : \mathcal{Y} \to \mathbb{R}$. At the end of a round, the learner has the option of purchasing the full loss function, again at a fixed prespecified cost $c_{d+1} \geq 0$ (by default, the loss function is not revealed to the learner). The learner's performance is measured by his regret as he competes against some prespecified set of predictors. Just like the learner, a competing predictor also needs to purchase the feature values needed in the prediction. If $s_t \in \{0,1\}^{d+1}$ is the indicator vector denoting what the learner purchased in round $t$ ($s_{t,i} = 1$ if the learner purchased $x_{t,i}$ for $1 \leq i \leq d$, and purchased the label for $i = d+1$) and $c \in [0,\infty)^{d+1}$ denotes the respective costs, then the regret with respect to a class of prediction functions $\mathcal{F} \subset \{f \mid f : \mathcal{X} \to \mathcal{Y}\}$ is defined by

$$R_T = \sum_{t=1}^T \left\{ \ell_t(\hat{y}_t) + \langle s_t, c \rangle \right\} \; - \; \inf_{f \in \mathcal{F}} \left\{ T \langle \mathrm{s}(f), c_{1:d} \rangle + \sum_{t=1}^T \ell_t(f(x_t)) \right\},$$

where $c_{1:d} \in \mathbb{R}^d$ is the vector obtained from $c$ by dropping its last component and for a given function $f : \mathbb{R}^d \to \mathcal{Y}$, $\mathrm{s}(f) \in \{0,1\}^d$ is an indicator vector whose $i^{\text{th}}$ component indicates whether $f$

is sensitive to its $i^{\text{th}}$ input (in particular, $\mathrm{s}_i(f) = 0$ by definition when $f(x_1, \ldots, x_i, \ldots, x_d) = f(x_1, \ldots, x_i', \ldots, x_d)$ holds for all $(x_1, \ldots, x_i, \ldots, x_d), (x_1, \ldots, x_i', \ldots, x_d) \in \mathcal{X}$; otherwise $\mathrm{s}_i(f) = 1$). Note that when defining the best competitor in hindsight, we did not include the cost of observing the loss function. This is because (i) the reference predictors do not need it; and (ii) if we did include the cost of observing the loss function for the reference predictors, then the loss of each predictor would just be increased by $c_{d+1}T$, and so the regret $R_T$ would just be reduced by $c_{d+1}T$, making it substantially easier for the learner to achieve sublinear regret. Thus, we prefer the current regret definition as it promotes the study of regret when there is a price attached to observing the loss functions.

To motivate our framework, consider the problem of developing a computer-assisted diagnostic tool to determine what treatment to apply to a patient in a subpopulation of patients. When a patient arrives, the computer can order a number of tests that cost money, while other information (e.g., the medical record of the patient) is available for free. Based on the available information, the system chooses a treatment. Following-up the patient may or may not incur additional cost. In this example, there is typically a delay in obtaining the information whether the treatment was effective. However, for simplicity, in this work we have decided not to study the effect of this delay. Several works in the literature show that delays usually increase the regret in a moderate fashion (Mesterharm, 2005; Weinberger and Ordentlich, 2006; Agarwal and Duchi, 2011; Joulani et al., 2013).

As another example, consider the problem of product testing in a manufacturing process (e.g., the production of electronic consumer devices). When the product arrives, it can be subjected to a large number of diagnostic tests that differ in terms of their costs and effectiveness. The goal is to predict whether the product is defect-free. Obtaining the ground truth can also be quite expensive, especially for complex products. The challenge is that the effectiveness of the various tests is often *a priori* unknown and that different tests may provide complementary information (meaning that many tests may be required). . Hence, it might be challenging to decide what form the most cost-effective diagnostic procedure may take. Yet another example is the problem of developing a cost-effective way of instrument calibration. In this problem, the goal is to predict one or more real-valued parameters of some product. Again, various tests with different costs and reliability can be used as the input to the predictor.

Finally, although we pose the task as an online learning problem, it is easy to show that the procedures we develop can also be used to attack the batch learning problem, when the goal is to learn a predictor that will be cost-efficient on future data given a database of examples.

Obviously, when observing the loss is costly, the problem is related to active learning. However, to our best knowledge, the case when observing the features is costly has not been studied before in the online learning literature. Section 1.1 will discusses the relationship of our work to the existing literature in more detail.

This paper analyzes two versions of the online problem. In the first version, *free-label online probing*, there is no cost to seeing the loss function, that is, $c_{d+1} = 0$. (The loss function often compares the predicted value with some label in a known way, in which case learning the value of the label for the round means that the whole loss function becomes known; hence the choice of the name.) Thus, the learner naturally will choose to see the loss function after he provides his prediction; this provides feedback that the learner can use, to improve the predictor he produces. In the second version, *non-free-label online probing*, the cost of seeing the loss function is positive: $c_{d+1} > 0$.

In Section 2 we study the case of free-label online probing. We give an algorithm that enjoys a regret of $\mathcal{O}(\sqrt{2^d LT \ln \mathcal{N}_T(1/(TL))})$ when the losses are $L$-equi-Lipschitz (Theorem 2.2), where $\mathcal{N}_T(\varepsilon)$ is the $\varepsilon$-covering number of $\mathcal{F}$ on sequences of length $T$. This leads to an $\tilde{\mathcal{O}}(\sqrt{2^d LT})$ regret bound for typical function classes, such as the class of linear predictors with bounded weights and bounded inputs. We also show that, in the worst case, the exponential dependence on the dimension cannot be avoided in the bound. For the special case of linear prediction with quadratic loss, we give an algorithm whose regret scales only as $\tilde{\mathcal{O}}(\sqrt{dt})$, a vast improvement in the dependence on $d$.

The case of non-free-label online probing is treated in Section 3. Here, in contrast to the free-label case, we prove that the minimax growth rate of the regret is of the order $\tilde{\Theta}(T^{2/3})$. The increase of regret-rate stems from the fact that the "best competitor in hindsight" does not have to pay for the label. In contrast to the previous case, since the label is costly here, if the algorithm decides to see the

label it does not even have to reason about which features to observe, as querying the label requires paying a cost that is a constant over the cost of the best predictor in hindsight, already resulting in the $\tilde{\Theta}(T^{2/3})$ regret rate. However, in practice (for shorter horizons) it still makes sense to select the features that provide the best balance between the feature-cost and the prediction loss. Although we do not study this, we note that by combining the algorithmic ideas developed for the free-label case with the ideas developed for the non-free-label case, it is possible to derive an algorithm that reasons actively about the cost of observing the features, too.

In the part dealing with the free-label problem, we build heavily on the results of Mannor and Shamir (2011), while in the part dealing with the non-free-label problem we build on the ideas of (Cesa-Bianchi et al., 2006). Due to space limitations, all of our proofs are relegated to the appendix.

## 1.1 Related Work

This paper analyzes online learning when features (and perhaps labels) have to be purchased. The standard "batch learning" framework has a pure explore phase, which gives the learner a set of labeled, completely specified examples, followed by a pure exploit phase, where the learned predictor is asked to predict the label for novel instances. Notice the learner is not required (nor even allowed) to decide which information to gather. By contrast, "active (batch) learning" requires the learner to identify that information (Settles, 2009). Most such active learners begin with completely specified, but unlabeled instances; they then purchase labels for a subset of the instances. Our model, however, requires the learner to purchase feature values as well. This is similar to the "active feature-purchasing learning" framework (Lizotte et al., 2003). This is extended in Kapoor and Greiner (2005) to a version that requires the eventual predictor (as well as the learner) to pay to see feature values as well. However, these are still in the batch framework: after gathering the information, the learner produces a predictor, which is not changed afterwards.

Our problem is an online problem over multiple rounds, where at each round the learner is required to predict the label for the current example. Standard online learning algorithms typically assume that each example is given with all the features. For example, Cesa-Bianchi et al. (2005) provided upper and lower bounds on the regret where the learner is given all the features for each example, but must pay for any labels he requests. In our problem, the learner must pay to see the values of the features of each example as well as the cost to obtain its true label at each round. This cost model means there is an advantage to finding a predictor that involves few features, as long as it is sufficiently accurate. The challenge, of course, is finding these relevant features, which happens during this online learning process.

Other works, in particular Rostamizadeh et al. (2011) and Dekel et al. (2010), assume the features of different examples might be corrupted, missed, or partially observed due to various problems, such as failure in sensors gathering these features. Having such missing features is realistic in many applications. Rostamizadeh et al. (2011) provided an algorithm for this task in the online settings, with optimal $\mathcal{O}(\sqrt{T})$ regret where $T$ is the number of rounds. Our model differs from this model as in our case the learner has the option to obtain the values of only the subset of the features that he selects.

## 2 Free-Label Probing

In this section we consider the case when the cost of observing the loss function is zero. Thus, we can assume without loss of generality that the learner receives the loss function at the end of each round (i.e., $s_{t,d+1} = 1$). We will first consider the general setting where the only restriction is that the losses are equi-Lipschitz and the function set $\mathcal{F}$ has a finite empirical worst-case covering number. Then we consider the special case where the set of competitors are the linear predictors and the losses are quadratic.

### 2.1 The Case of Lipschitz losses

In this section we assume that the loss functions, $\ell_t$, are Lipschitz with a known, common Lipschitz constant $L$ over $\mathcal{Y}$ w.r.t. to some semi-metric $d_{\mathcal{Y}}$ of $\mathcal{Y}$: for all $t \geq 1$

$$\sup_{y,y' \in \mathcal{Y}} |\ell_t(y) - \ell_t(y')| \leq L \, d_{\mathcal{Y}}(y, y'). \tag{1}$$

Clearly, the problem is an instance of prediction with expert advice under partial information feedback (Auer et al., 2002), where each expert corresponds to an element of $\mathcal{F}$. Note that, if the learner chooses to observe the values of some features, then he will also be able to evaluate the losses of all the predictors $f \in \mathcal{F}$ that use only these selected features. This can be formalized as follows: By a slight abuse of notation let $s_t \in \{0,1\}^d$ be the indicator showing the features selected by the learner at time $t$ (here we drop the last element of $s_t$ as $s_{t,d_1}$ is always 1); similarly, we will drop the last coordinate of the cost vector $c$ throughout this section. Then, the learner can compute the loss of any predictor $f \in \mathcal{F}$ such that $\mathrm{s}(f) \leq s_t$, where $\leq$ denotes the conjunction of the component-wise comparison. However, for some loss functions, it may be possible to estimate the losses of other predictors, too. We will exploit this when we study some interesting special cases of the general problem. However, in general, it is not possible to infer the losses for functions such that $s_{t,i} < \mathrm{s}(f)_i$ for some $i$ (cf. Theorem 2.3).

The idea is to study first the case when $\mathcal{F}$ is finite and then reduce the general case to the finite case by considering appropriate finite coverings of the space $\mathcal{F}$. The regret will then depend on how the covering numbers of the space $\mathcal{F}$ behave.

Mannor and Shamir (2011) studied problems similar to this in a general framework, where in addition to the loss of the selected predictor (expert), the losses of some other predictors are also communicated to the learner in every round. The connection between the predictors is represented by a directed graph whose nodes are labeled as elements of $\mathcal{F}$ (i.e., as the experts) and there is an edge from $f \in \mathcal{F}$ to $g \in \mathcal{F}$ if, when choosing $f$, the loss of $g$ is also revealed to the learner. It is assumed that the graph of any round $t$, $G_t = (\mathcal{F}, E_t)$ becomes known to the learner at the beginning of the round. Further, it is also assumed that $(f, f) \in E_t$ for every $t \geq 1$ and $f \in \mathcal{F}$. Mannor and Shamir (2011) gave an algorithm, called ELP (exponential weights with linear programming), to solve this problem, which calls the Exponential Weights algorithm, but modifies it to explore less, exploiting the information structure of the problem. The exploration distribution is found by solving a linear program, explaining the name of the algorithm. The regret of ELP is analyzed in the following theorem.

**Theorem 2.1** (Mannor and Shamir 2011). *Consider a prediction with expert advice problem over $\mathcal{F}$ where in round $t$, $G_t = (\mathcal{F}, E_t)$ is the directed graph that encodes which losses become available to the learner. Assume that for any $t \geq 1$, at most $\chi(G_t)$ cliques of $G_t$ can cover all vertices of $G_t$. Let $B$ be a bound on the non-negative losses $\ell_t$: $\max_{t \geq 1, f \in \mathcal{F}} \ell_t(f(x_t)) \leq B$. Then, there exists a constant $C_{\mathrm{ELP}} > 0$ such that for any $T > 0$, the regret of Algorithm 2 (shown in the Appendix) when competing against the best predictor using* ELP *satisfies*

$$\mathbb{E}[R_T] \quad \leq \quad C_{\mathrm{ELP}} B \sqrt{(\ln |\mathcal{F}|) \sum_{t=1}^{T} \chi(G_t)}. \tag{2}$$

*The algorithm's computational cost in any given round is* $\mathrm{poly}(|\mathcal{F}|)$.

For a finite $\mathcal{F}$, define $E_t \equiv E \doteq \{(f,g) \mid \mathrm{s}(g) \leq \mathrm{s}(f)\}$. Then clearly, $\chi(G_t) \leq 2^d$. Further, $B = \|c_{1:d}\|_1 + \max_{t \geq 1, y \in \mathcal{Y}} \ell_t(y) \doteq C_1 + \ell_{\max}$ (i.e., $C_1 = \|c_{1:d}\|_1$). Plugging these into (2) gives

$$\mathbb{E}[R_T] \quad \leq \quad C_{\mathrm{ELP}}(C_1 + \ell_{\max}) \sqrt{2^d T \ln |\mathcal{F}|}. \tag{3}$$

To apply this algorithm in the case when $\mathcal{F}$ is infinite, we have to approximate $\mathcal{F}$ with a finite set $\mathcal{F}' \subset \{f \mid f : X \to \mathcal{Y}\}$. The worst-case maximum approximation error of $\mathcal{F}$ using $\mathcal{F}'$ over sequences of length $T$ can be defined as

$$A_T(\mathcal{F}', \mathcal{F}) = \max_{x \in \mathcal{X}^T} \sup_{f \in \mathcal{F}} \inf_{f' \in \mathcal{F}'} \frac{1}{T} \sum_{t=1}^{T} d_{\mathcal{Y}}(f(x_t), f'(x_t)) + \langle (\mathrm{s}(f') - \mathrm{s}(f))^+, c_{1:d} \rangle,$$

where $(\mathrm{s}(f') - \mathrm{s}(f))^+$ denotes the coordinate-wise positive part of $\mathrm{s}(f') - \mathrm{s}(f)$, that is, the indicator vector of the features used by $f'$ and not used by $f$. The average error can also be viewed as a (normalized) $d_{\mathcal{Y}}$-"distance" between the vectors $(f(x_t))_{1 \leq t \leq T}$ and $(f'(x_t))_{1 \leq t \leq T}$ penalized with the extra feature costs. For a given positive number $\alpha$, define the *worst-case empirical covering number* of $\mathcal{F}$ at level $\alpha$ and horizon $T > 0$ by

$$\mathcal{N}_T(\mathcal{F}, \alpha) = \min\{ |\mathcal{F}'| \mid \mathcal{F}' \subset \{f \mid f : X \to \mathcal{Y}\}, A_T(\mathcal{F}', \mathcal{F}) \leq \alpha \}.$$

We are going to apply the ELP algorithm to $\mathcal{F}'$ and apply (3) to obtain a regret bound. If $f'$ uses more features than $f$ then the cost-penalized distance between $f'$ and $f$ is bounded from below by the cost of observing the extra features. This means that unless the problem is very special, $\mathcal{F}'$ has to contain, for all $s \in \{s(f) \mid f \in \mathcal{F}\}$, some $f'$ with $s(f') = s$. Thus, if $\mathcal{F}$ contains a function for all $s \in \{0,1\}^d$, $\chi(G_t) = 2^d$. Selecting a covering $\mathcal{F}'$ that achieves accuracy $\alpha$, the approximation error becomes $TL\alpha$ (using equation 1), giving the following bound:

**Theorem 2.2.** *Assume that the losses $(\ell_t)_{t\geq 1}$ are $L$-Lipschitz (cf. (1)) and $\alpha > 0$. Then, there exists an algorithm such that for any $T > 0$, knowing $T$, the regret satisfies*

$$\mathbb{E}[R_T] \leq C_{\mathrm{ELP}}(C_1 + \ell_{\max})\sqrt{2^d T \ln \mathcal{N}_T(\mathcal{F}, \alpha)} + TL\alpha.$$

*In particular, by choosing $\alpha = 1/(TL)$, we have*

$$\mathbb{E}[R_T] \leq C_{\mathrm{ELP}}(C_1 + \ell_{\max})\sqrt{2^d T \ln \mathcal{N}_T(\mathcal{F}, 1/(TL))} + 1.$$

We note in passing that the the dependence of the algorithm on the time horizon $T$ can be alleviated, using, for example, the doubling trick.

In order to turn the above bound into a concrete bound, one must investigate the behavior of the *metric entropy*, $\ln \mathcal{N}_T(\mathcal{F}, \alpha)$. In many cases, the metric entropy can be bounded independently of $T$. In fact, often, $\ln \mathcal{N}_T(\mathcal{F}, \alpha) = D \ln(1 + c/\alpha)$ for some $c, D > 0$. When this holds, $D$ is often called the "dimension" of $\mathcal{F}$ and we get that

$$\mathbb{E}[R_T] \leq C_{\mathrm{ELP}}(C_1 + \ell_{\max})\sqrt{2^d T D \ln(1 + cTL)} + 1.$$

As a specific example, we will consider the case of real-valued linear functions over a ball in a Euclidean space with weights belonging to some other ball. For a normed vector space $V$ with norm $\|\cdot\|$ and dual norm $\|\cdot\|_*$, $x \in V$, $r \geq 0$, let $B_{\|\cdot\|}(x,r) = \{v \in V \mid \|v\| \leq r\}$ denote the ball in $V$ centered at $x$ that has radius $r$. For $\mathcal{X} \subset \mathbb{R}^d$, $\mathcal{W} \subset \mathbb{R}^d$, let

$$\mathcal{F} \subset \mathrm{Lin}(\mathcal{X}, \mathcal{W}) \doteq \{g : \mathcal{X} \to \mathbb{R} \mid g(\cdot) = \langle w, \cdot \rangle, w \in \mathcal{W}\} \tag{4}$$

be the space of linear mappings from $\mathcal{X}$ to reals with weights belonging to $\mathcal{W}$. We have the following lemma:

**Lemma 2.1.** *Let $X, W > 0$, $d_{\mathcal{Y}}(y, y') = |y - y'|$, $\mathcal{X} \subset B_{\|\cdot\|}(0, X)$ and $\mathcal{W} \subset B_{\|\cdot\|_*}(0, W)$. Consider a set of real-valued linear predictors $\mathcal{F} \subset \mathrm{Lin}(\mathcal{X}, \mathcal{W})$. Then, for any $\alpha > 0$,*

$$\ln \mathcal{N}_T(\mathcal{F}, \alpha) \leq d \ln(1 + 2WX/\alpha).$$

The previous lemma, together with Theorem 2.2 immediately gives the following result:

**Corollary 2.1.** *Assume that $\mathcal{F} \subset \mathrm{Lin}(\mathcal{X}, \mathcal{W})$, $\mathcal{X} \subset B_{\|\cdot\|}(0, X)$, $\mathcal{W} \subset B_{\|\cdot\|_*}(0, W)$ for some $X, W > 0$. Further, assume that the losses $(\ell_t)_{t\geq 1}$ are $L$-Lipschitz. Then, there exists an algorithm such that for any $T > 0$, the regret of the algorithm satisfies,*

$$\mathbb{E}[R_T] \leq C_{\mathrm{ELP}}(C_1 + \ell_{\max})\sqrt{d 2^d T \ln(1 + 2TLWX)} + 1.$$

Note that if one is given an *a priori bound* $p$ on the maximum number of features that can be used in a single round (allowing the algorithm to use fewer than $p$, but not more features) then $2^d$ in the above bound could be replaced by $\sum_{1 \leq i \leq p} \binom{d}{i} \approx d^p$, where the approximation assumes that $p < d/2$. Such a bound on the number of features available per round may arise from strict budgetary considerations. When $d^p$ is small, this makes the bound non-vacuous even for small horizons $T$. In addition, in such cases the algorithm also becomes computationally feasible. It remains an interesting open question to study the computational complexity when there is no restriction on the number of features used. In the next theorem, however, we show that the worst-case exponential dependence of the regret on the number of features cannot be improved (while keeping the root-$T$ dependence on the horizon). The bound is based on the lower bound construction of Mannor and Shamir (2011), which reduces the problem to known lower bounds in the multi-armed bandit case.

**Theorem 2.3.** *There exist an instance of* free-label online probing *such that the minimax regret of any algorithm is* $\Omega\left(\sqrt{\binom{d}{d/2} T}\right)$.

## 2.2 Linear Prediction with Quadratic Losses

In this section, we study the problem under the assumption that the predictors have a linear form and the loss functions are quadratic. That is, $\mathcal{F} \subset \mathrm{Lin}(\mathcal{X}, \mathcal{W})$ where $\mathcal{W} = \{w \in \mathbb{R}^d \mid \|w\|_* \leq w_{\mathrm{lim}}\}$ and $\mathcal{X} = \{x \in \mathbb{R}^d \mid \|x\| \leq x_{\mathrm{lim}}\}$ for some given constants $w_{\mathrm{lim}}, x_{\mathrm{lim}} > 0$, while $\ell_t(y) = (y - y_t)^2$, where $|y_t| \leq x_{\mathrm{lim}} w_{\mathrm{lim}}$. Thus, choosing a predictor is akin to selecting a weight vector $w_t \in \mathcal{W}$, as well as a binary vector $s_t \in \mathcal{G} \subset \{0,1\}^d$ that encodes the features to be used in round $t$. The prediction for round $t$ is then $\hat{y}_t = \langle w_t, s_t \odot x_t \rangle$, where $\odot$ denotes coordinate-wise product, while the loss suffered is $(\hat{y}_t - y_t)^2$. The set $\mathcal{G}$ is an arbitrary non-empty, a priori specified subset of $\{0,1\}^d$ that allows the user of the algorithm to encode extra constraints on what subsets of features can be selected.

In this section we show that in this case a regret bound of size $\tilde{\mathcal{O}}(\sqrt{\mathrm{poly}(d)T})$ is possible. The key idea that permits the improvement of the regret bound is that a randomized choice of a weight vector $W_t$ (and thus, of a subset) helps one construct unbiased estimates of the losses $\ell_t(\langle w, s \odot x_t \rangle)$ for all weight vectors $w$ and all subsets $s \in \mathcal{G}$ under some mild conditions on the distribution of $W_t$. That the construction of such unbiased estimates is possible, despite that some feature values are unobserved, is because of the special algebraic structure of the prediction and loss functions. A similar construction has appeared in a different context, e.g., in the paper of Cesa-Bianchi et al. (2010).

The construction works as follows. Define the $d \times d$ matrix, $X_t$ by $(X_t)_{i,j} = x_{t,i} x_{t,j}$ ($1 \leq i, j \leq d$). Expanding the loss of the prediction $\hat{y}_t = \langle w, x_t \rangle$, we get that the loss of using $w \in \mathcal{W}$ is

$$\ell_t(w) \doteq \ell_t(\langle w, x_t \rangle) = w^\top X_t w - 2 w^\top x_t y_t + y_t^2,$$

where with a slight abuse of notation we have introduced the loss function $\ell_t : \mathcal{W} \to \mathbb{R}$ (we'll keep abusing the use of $\ell_t$ by overloading it based on the type of its argument). Clearly, it suffices to construct unbiased estimates of $\ell_t(w)$ for any $w \in \mathcal{W}$.

We will use a discretization approach. Therefore, assume that we are given a finite subset $\mathcal{W}'$ of $\mathcal{W}$ that will be constructed later. In each step $t$, our algorithm will choose a random weight vector $W_t$ from a probability distribution supported on $\mathcal{W}'$. Let $p_t(w)$ be the probability of selecting the weight vector, $w \in \mathcal{W}'$. For $1 \leq i \leq d$, let

$$q_t(i) = \sum_{w \in \mathcal{W}' : i \in s(w)} p_t(w),$$

be the probability that $s(W_t)$ will contain $i$, while for $1 \leq i, j \leq d$, let

$$q_t(i,j) = \sum_{w \in \mathcal{W}' : i,j \in s(w)} p_t(w),$$

be the probability that both $i, j \in s(W_t)$.[2] Assume that $p_t(\cdot)$ is constructed such that $q_t(i,j) > 0$ holds for any time $t$ and indices $1 \leq i, j \leq d$. This also implies that $q_t(i) > 0$ for all $1 \leq i \leq d$. Define the vector $\tilde{x}_t \in \mathbb{R}^d$ and matrix $\tilde{X}_t \in \mathbb{R}^{d \times d}$ using the following equations:

$$\tilde{x}_{t,i} = \frac{\mathbb{1}_{\{i \in s(W_t)\}} x_{t,i}}{q_t(i)}, \qquad (\tilde{X}_t)_{i,j} = \frac{\mathbb{1}_{\{i,j \in s(W_t)\}} x_{t,i} x_{t,j}}{q_t(i,j)}. \tag{5}$$

It can be readily verified that $\mathbb{E}[\tilde{x}_t \mid p_t] = x_t$ and $\mathbb{E}\left[\tilde{X}_t \mid p_t\right] = X_t$. Further, notice that both $\tilde{x}_t$ and $\tilde{X}_t$ can be computed based on the information available at the end of round $t$, i.e., based on the feature values $(x_{t,i})_{i \in s(W_t)}$. Now, define the estimate of prediction loss

$$\tilde{\ell}_t(w) = w^\top \tilde{X}_t w - 2 w^\top \tilde{x}_t y_t + y_t^2. \tag{6}$$

Note that $y_t$ can be readily computed from $\ell_t(\cdot)$, which is available to the algorithm (equivalently, we may assume that the algorithm observed $y_t$). Due to the linearity of expectation, we have $\mathbb{E}\left[\tilde{\ell}_t(w) | p_t\right] = \ell_t(w)$. That is, $\tilde{\ell}_t(w)$ provides an unbiased estimate of the loss $\ell_t(w)$ for any $w \in \mathcal{W}$. Hence, by adding a feature cost term we get $\tilde{\ell}_t(w) + \langle s(w), c \rangle$ as an estimate of the loss that the learner would have suffered at round $t$ had he chosen the weight vector $w$.

**Algorithm 1** The LQDEXP3 Algorithm

---

**Parameters:** Real numbers $0 \leq \eta$, $0 < \gamma \leq 1$, $\mathcal{W}' \subset \mathcal{W}$ finite set, a distribution $\mu$ over $\mathcal{W}'$, horizon $T > 0$.
**Initialization:** $u_1(w) = 1$ $(w \in \mathcal{W}')$.
**for** $t = 1$ **to** $T$ **do**
  Draw $W_t \in \mathcal{W}'$ from the probability mass function

$$p_t(w) = (1 - \gamma)\frac{u_t(w)}{U_t} + \gamma\mu(w), \qquad w \in \mathcal{W}'.$$

  Obtain the features values, $(x_{t,i})_{i \in s(W_t)}$.
  Predict $\hat{y}_t = \sum_{i \in s(W_t)} w_{t,i} x_{t,i}$.
  **for** $w \in \mathcal{W}'$ **do**
    Update the weights using (6) for the definitions of $\tilde{\ell}_t(w)$:

$$u_{t+1}(w) = u_t(w)e^{-\eta(\tilde{\ell}_t(w) + \langle c, s(w) \rangle)}, \quad w \in \mathcal{W}'.$$

  **end for**
**end for**

---

### 2.2.1 LQDExp3 – A Discretization-based Algorithm

Next we show that the standard EXP3 Algorithm applied to a discretization of the weight space $\mathcal{W}$ achieves $\mathcal{O}(\sqrt{dT})$ regret. The algorithm, called LQDEXP3 is given as Algorithm 1. In the name of the algorithm, LQ stands for linear prediction with quadratic losses and D denotes discretization. Note that if the exploration distribution $\mu$ in the algorithm is such that for any $1 \leq i, j \leq d$, $\sum_{w \in W': i, j \in s(w)} \mu(w) > 0$ then $q_t(i, j) > 0$ will be guaranteed for all time steps. Using the notation $y_{\lim} = w_{\lim}x_{\lim}$ and $E_{\mathcal{G}} = \max_{s \in \mathcal{G}} \sup_{w \in \mathcal{W}: \|w\|_* = 1} \|w \odot s\|_*$, we can state the following regret bound on the algorithm

**Theorem 2.4.** *Let $w_{\lim}, x_{\lim} > 0$, $c \in [0, \infty)^d$ be given, $\mathcal{W} \subset B_{\|\cdot\|_*}(0, w_{\lim})$ convex, $\mathcal{X} \subset B_{\|\cdot\|}(0, x_{\lim})$ and fix $T \geq 1$. Then, there exist a parameter setting for* LQDEXP3 *such that the following holds: Let $R_T$ denote the regret of* LQDEXP3 *against the best linear predictor from* $\mathrm{Lin}(\mathcal{W}, \mathcal{X})$ *when* LQDEXP3 *is used in an online free-label probing problem defined with the sequence $((x_t, y_t))_{1 \leq t \leq T}$ ($\|x_t\| \leq x_{\lim}$, $|y_t| \leq y_{\lim}$, $1 \leq t \leq T$), quadratic losses $\ell_t(y) = (y - y_t)^2$, and feature-costs given by the vector c. Then,*

$$\mathbb{E}[R_T] \leq C \sqrt{Td\left(4y_{\lim}^2 + \|c\|_1\right)(w_{\lim}^2 x_{\lim}^2 + 2y_{\lim}w_{\lim}x_{\lim} + 4y_{\lim}^2 + \|c\|_1)\ln(E_{\mathcal{G}}T)},$$

*where $C > 0$ is a universal constant (i.e., the value of $C$ does not depend on the problem parameters).*

The actual parameter setting to be used with the algorithm is constructed in the proof. The computational complexity of LQDEXP3 is exponential in the dimension $d$ due to the discretization step, hence quickly becomes impractical when the number of features is large. On the other hand, one can easily modify the algorithm to run without discretization by replacing EXP3 with its continuous version. The resulting algorithm enjoys essentially the same regret bound, and can be implemented efficiently whenever efficient sampling is possible from the resulting distribution. This approach seems to be appealing, since, from a first look, it seems to involve sampling from truncated Gaussian distributions, which can be done efficiently. However, it is easy to see that when the sampling probabilities of some feature are small, the estimated loss will not be convex as $\tilde{X}_t$ may not be positive semi-definite, and therefore the resulting distributions will not always be truncated Gaussians. Finding an efficient sampling procedure for such situations is an interesting open problem.

The optimality of LQDEXP3 can be seen by the following lower bound on the regret:

**Theorem 2.5.** *Let $d > 0$, and consider the online free label probing problem with linear predictors, where $\mathcal{W} = \{w \in \mathbb{R}^d \mid \|w\|_1 \leq w_{\lim}\}$ and $\mathcal{X} = \{x \in \mathbb{R}^d \mid \|x\|_\infty \leq 1\}$. Assume, for all $t \geq 1$, that the loss functions are of the form $\ell_t(w) = (w^\top x_t - y_t)^2 + \langle s(w), c \rangle$, where $|y_t| \leq 1$ and $c = 1/2 \times \mathbf{1} \in \mathbb{R}^d$. Then, for any prediction algorithm and for any $T \geq \frac{4d}{8\ln(4/3)}$, there exists a*

*sequence $((x_t, y_t))_{1 \leq t \leq T} \in (\mathcal{X} \times [-1,1])^T$ such that the regret of the algorithm can be bounded from below as*

$$\mathbb{E}[R_T] \geq \frac{\sqrt{2}-1}{\sqrt{32 \ln(4/3)}} \sqrt{Td}.$$

## 3 Non-Free-Label Probing

If $c_{d+1} > 0$, the learner has to pay for observing the true label. This scenario is very similar to the well-known label-efficient prediction case in online learning (Cesa-Bianchi et al., 2006). In fact, the latter problem is a special case of this problem, immediately giving us that the regret of any algorithm is at least of order $T^{2/3}$. It turns out that if one observes the (costly) label in a given round then it does not effect the regret rate if one observes all the features at the same time. The resulting "revealing action algorithm", given in Algorithm 3 in the Appendix, achieves the following regret bound for finite expert classes:

**Lemma 3.1.** *Given any non-free-label online probing with finitely many experts, Algorithm 3 with appropriately set parameters achieves*

$$\mathbb{E}[R_T] \leq C \max \left( T^{2/3} (\ell_{\max}^2 \|c\|_1 \ln |\mathcal{F}|)^{1/3}, \ell_{\max} \sqrt{T \ln |\mathcal{F}|} \right)$$

*for some constant $C > 0$.*

Using the fact that, in the linear prediction case, approximately $(2TLWX + 1)^d$ experts are needed to approximate each expert in $\mathcal{W}$ with precision $\alpha = \frac{1}{LT}$ in worst-case empirical covering, we obtain the following theorem (note, however, that the complexity of the algorithm is again exponential in the dimension $d$, as we need to keep a weight for each expert):

**Theorem 3.1.** *Given any non-free-label online probing with linear predictor experts and Lipschitz prediction loss function with constant L, Algorithm 3 with appropriately set parameters running on a sufficiently discretized predictor set achieves*

$$\mathbb{E}[R_T] \leq C \max \left( T^{2/3} \left[ \ell_{\max}^2 \|c\|_1 \, d \, \ln(TLWX) \right]^{1/3}, \ell_{\max} \sqrt{Td \ln(TLWX)} \right)$$

*for some universal constant $C > 0$.*

That Algorithm 3 is essentially optimal for linear predictions and quadratic losses is a consequence of the following almost matching lower bound:

**Theorem 3.2.** *There exists a constant $C$ such that, for any non-free-label probing with linear predictors, quadratic loss, and $c_j > (1/d) \sum_{i=1}^{d} c_i - 1/2d$ for every $j = 1, \ldots, d$, the expected regret of any algorithm can be lower bounded by*

$$\mathbb{E}[R_T] \geq C(c_{d+1}d)^{1/3}T^{2/3}.$$

## 4 Conclusions

We introduced a new problem called *online probing*. In this problem, the learner has the option of choosing the subset of features he wants to observe as well as the option of observing the true label, but has to pay for this information. This setup produced new challenges in solving the online problem. We showed that when the labels are free, it is possible to devise algorithms with optimal regret rate $\Theta(\sqrt{T})$ (up to logarithmic factors), while in the non-free-label case we showed that only $\Theta(T^{2/3})$ is achievable. We gave algorithms that achieve the optimal regret rate (up to logarithmic factors) when the number of experts is finite or in the case of linear prediction. Unfortunately either our bounds or the computational complexity of the corresponding algorithms are exponential in the problem dimension, and it is an open problem whether these disadvantages can be eliminated simultaneously.

## Acknowledgements

The authors thank Yevgeny Seldin for finding a bug in an earlier version of the paper. This work was supported in part by DARPA grant MSEE FA8650-11-1-7156, the Alberta Innovates Technology Futures, AICML, and the Natural Sciences and Engineering Research Council (NSERC) of Canada.

## Footnotes

[1] We use $^\top$ to denote the transpose of vectors. Throughout, all vectors $x \in \mathbb{R}^d$ will denote column vectors.

[2] Note that, following our earlier suggestion, we view the $d$-dimensional binary vectors as subsets of $\{1, \ldots, d\}$.

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
