[Supplementary Material]

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

# APPENDIX—SUPPLEMENTARY MATERIAL

## A.1 Free-Label Probing: Lipschitz losses

### A.1.1 The ELP algorithm of Mannor and Shamir (2011)

---

**Algorithm 2** The ELP Algorithm. In the pseudocode, $\Delta_N$ denotes the $N$-dimensional simplex: $\Delta_N = \{s \in [0,1]^N \mid \sum_{i=1}^N s_i = 1\}$.

---

**Parameters:** Neighborhood graphs $G_t = (\mathcal{F}, E_t)$, $1 \leq t \leq T$, a bound $B$ on the losses.
**Initialization:** $N = |\mathcal{F}|$, $\beta = \sqrt{(\ln N)/(3B^2 \sum_t \chi(G_t))}$, $w_{0,j} = 1/N$, $1 \leq j \leq N$.
**for** $t = 1$ **to** $T$ **do**
   Let $s_t = \arg\max_{q \in \Delta_N} \min_{1 \leq i \leq N} \sum_{(i,k) \in E_t} q_k$.
   Let $s_t^* = \min_{1 \leq i \leq N} \sum_{(i,k) \in E_t} s_{t,i}$.
   Let $\gamma_t = \beta B / s_t^*$.
   Choose action $i_t$ randomly from probability mass function
$$p_{t,i} = (1 - \gamma_t) \frac{w_{t,i}}{\sum_{j=1}^N w_{t,j}} + \gamma_t s_{t,i} \qquad (1 \leq i \leq N).$$

   Receive loss $(\ell_{t,k})_{(i_t,k) \in E_t}$.
   Compute $\tilde{g}_{t,j} = \frac{B - \ell_{t,j}}{\sum_{(l,j) \in E_t} p_{t,l}}$ if $(j, i_t) \in E_t$, and $\tilde{g}_j(t) = 0$ otherwise.
   $w_{t+1,j} = w_{t,j} \exp(\beta \tilde{g}_{t,j})$, $1 \leq j \leq N$.
**end for**

---

### A.1.2 Proofs

**Lemma 2.1.** *Let* $X, W > 0$, $d_{\mathcal{Y}}(y, y') = |y - y'|$, $\mathcal{X} \subset B_{\|\cdot\|}(0, X)$ *and* $\mathcal{W} \subset B_{\|\cdot\|_*}(0, W)$. *Consider a set of real-valued linear predictors* $\mathcal{F} \subset \mathrm{Lin}(\mathcal{X}, \mathcal{W})$. *Then, for any* $\alpha > 0$,
$$\ln \mathcal{N}_T(\mathcal{F}, \alpha) \leq d \ln(1 + 2WX/\alpha).$$

*Proof.* An appropriate covering of $\mathcal{F}$ can be constructed as follows: Consider an $\varepsilon$-covering $\mathcal{W}'$ of the ball $\mathcal{W}$ with respect to $\|\cdot\|_*$ for some $\varepsilon > 0$ (*i.e.,* for any $w \in \mathcal{W}$ there exists $w' \in \mathcal{W}'$ such that $\|w - w'\|_* \leq \varepsilon$). Then,
$$\mathcal{F}' = \{g : \mathcal{X} \to \mathbb{R} \mid g(x) = \langle x, w \rangle, w \in \mathcal{W}'\} \tag{7}$$
is an $\varepsilon X$-covering of $\mathcal{F}$. To see this pick any $f \in \mathcal{F}$. Thus, $f(x) = \langle w, x \rangle$ for some $w \in \mathcal{W}$. Take the vector in $\mathcal{W}'$ that is closest to $w$ and call it $w'$. Thus, $\|w - w'\|_* \leq \varepsilon$. Let $g \in \mathcal{F}'$ be given by $g(x) = \langle x, w' \rangle$. Then,
$$\frac{1}{T} \sum_{t=1}^T |f(x_t) - g(x_t)| = \frac{1}{T} \sum_{t=1}^T |\langle w - w', x_t \rangle| \leq \varepsilon X, \tag{8}$$
where in the last step we used Hölder's inequality and that by assumption $x_t \in \mathcal{X}$ and thus $\|x_t\| \leq X$. This argument thus shows that to get an $\alpha$-covering of $\mathcal{F}$, we need an $\varepsilon$-covering of $\mathcal{W}$ with $\varepsilon = \alpha/X$ and therefore $\mathcal{N}_T(\mathcal{F}, \alpha) \leq \mathcal{N}(\mathcal{W}, \alpha/X)$. As it is well known, $\mathcal{N}(\mathcal{W}, \varepsilon) \leq (2W/\varepsilon + 1)^d$ and thus $\ln \mathcal{N}_T(\mathcal{F}, \alpha) \leq d \ln(1 + 2WX/\alpha)$. $\square$

**Theorem 2.3.** *There exist an instance of* free-label online probing *such that the minimax regret of any algorithm is* $\Omega\left(\sqrt{\binom{d}{d/2} T}\right)$.

*Proof.* Let $\mathcal{X} = \{0,1\}^d$ and let $\mathcal{F}$ contain the XOR functions applied to all possible subsets of features: $\mathcal{F} = \{\mathrm{XOR}_S \mid S \subset \{1, \ldots, d\}\}$, where $\mathrm{XOR}_S(x) = \otimes_{i \in S} x_i$ ($\otimes$ denotes the Boolean XOR operator). Let the input vector $x_t \in \mathcal{X}$ at time $t$ be such that its components are generated

independently from each other from the uniform distribution and let them be independent of inputs generated at different time indices. The prediction space is restricted to $\mathcal{Y} = \{0, 1\}$ and the loss is defined to be the zero-one loss: $\ell_t(y) = \mathbb{1}_{\{y \neq \mathrm{XOR}_{S^*}(x_t)\}}$, where $S^* \subset \{1, \ldots, d\}$. The cost of the individual features is uniform. In particular, $c_i = 1/(2d)$.

Note that if an algorithm chooses *not* to observe some feature $i \in \{1, \ldots, d\}$ at some time step $t$, there is now way the algorithm can find out the result of $\mathrm{XOR}_S(x_t)$ for any $S$ containing $i$.[3] Hence, no algorithm can infer the losses for such functions. It is also clear that if the feature values for some set of features $S$ are observed then the loss for any function $\mathrm{XOR}_{S'}$ with $S' \subset S$ can be inferred from the observed loss function. Thus, the directed graph over $\mathcal{F}$ that connects $f \in \mathcal{F}$ to $f' \in \mathcal{F}$ when given the loss for $f$, one also learns the loss of $f'$, is isomorphic to the graph obtained from the lattice structure of subsets of $\{1, \ldots, d\}$. This latter graph has an independent set of size $\binom{d}{d/2}$ (*i.e.,* the set of all functions that use exactly $d/2$ features) and thus we can apply the same method that Mannor and Shamir (2011) uses to prove their Theorem 4 to get the desired lower bound for this problem. □

## A.2 Free Label Probing: Linear Prediction

### A.2.1 Upper Bound on the Regret

**Theorem 2.4.** *Let $w_{\lim}, x_{\lim} > 0$, $c \in [0, \infty)^d$ be given, $\mathcal{W} \subset B_{\|\cdot\|_*}(0, w_{\lim})$ convex, $\mathcal{X} \subset B_{\|\cdot\|}(0, x_{\lim})$ and fix $T \geq 1$. Then, there exist a parameter setting for $\mathrm{LQDEXP3}$ such that the following holds: Let $R_T$ denote the regret of $\mathrm{LQDEXP3}$ against the best linear predictor from $\mathrm{Lin}(\mathcal{W}, \mathcal{X})$ when $\mathrm{LQDEXP3}$ is used in an online free-label probing problem defined with the sequence $((x_t, y_t))_{1 \leq t \leq T}$ ($\|x_t\| \leq x_{\lim}$, $|y_t| \leq y_{\lim}$, $1 \leq t \leq T$), quadratic losses $\ell_t(y) = (y - y_t)^2$, and feature-costs given by the vector c. Then,*

$$\mathbb{E}[R_T] \leq C \sqrt{Td\left(4y_{\lim}^2 + \|c\|_1\right)(w_{\lim}^2 x_{\lim}^2 + 2y_{\lim}w_{\lim}x_{\lim} + 4y_{\lim}^2 + \|c\|_1)\ln(E_{\mathcal{G}}T)},$$

*where $C > 0$ is a universal constant (i.e., the value of $C$ does not depend on the problem parameters).*

Before stating the proof, we state a lemma that will be needed in the proof of this theorem. The lemma gives a bound on the regret of an exponential weights algorithm as a function of some "statistics" of the losses fed to the algorithm. Since the result is essentially extracted from the paper by Auer et al. (2002), its proof is omitted.

**Lemma A.2.1.** *Fix the integers $N, T > 0$, the real numbers $0 < \gamma < 1$, $\eta > 0$ and let $\mu$ be a probability mass function over the set $\underline{N} = \{1, \ldots, N\}$. Let $\ell_t : \underline{N} \to \mathbb{R}$ be a sequence of loss functions such that*

$$\eta\ell_t(i) \geq -1 \tag{9}$$

*holds for all $1 \leq t \leq T$ and $i \in \underline{N}$. Define the sequence of functions $(u_t)_{1 \leq t \leq T}$, $(p_t)_{1 \leq t \leq T}$ ($u_t : \underline{N} \to \mathbb{R}^+$, $p_t : \underline{N} \to [0, 1]$) by $u_t \equiv 1$,*

$$u_t(i) = \exp\left(\eta \sum_{s=1}^{t-1} \ell_s(i)\right), \quad p_t(i) = (1 - \gamma)\frac{u_t(i)}{\sum_{j \in \underline{N}} u_t(j)} + \gamma\mu(i), \quad (i \in \underline{N}, 1 \leq t \leq T+1).$$

*Let $\hat{L}_T = \sum_{t=1}^{T} \sum_{j \in \underline{N}} p_t(j)\ell_t(j)$ and $L_T(i) = \sum_{t=1}^{T} \ell_t(i)$. Then, for any $i \in \underline{N}$,*

$$\hat{L}_T - L_T(i) \leq \frac{\ln N}{\eta} + \eta \sum_{t=1}^{T} \sum_{j \in \underline{N}} p_t(j)\ell_t^2(j) + \gamma \sum_{t=1}^{T} \sum_{j \in \underline{N}} \mu(j)\{\ell_t(j) - \ell_t(i)\}.$$

*Proof of Theorem 2.4.* Fix the sequence of $((x_t, y_t))_{1 \leq t \leq T}$ as in the statement of the theorem and let $\ell_t(y) = (y - y_t)^2$. Remember that (with a slight abuse of notation), the loss of using weight $w \in \mathcal{W}$ in time step $t$ is

$$\ell_t(w) = \ell_t(\langle w, x_t \rangle), \quad 1 \leq t \leq T.$$

Now, observe that $\hat{y}_t = \sum_{i \in s(W_t)} W_{t,i} x_{t,i} = \langle W_t, x_t \rangle$ holds thanks to the definition of $s(W_t)$ and thus $\ell_t(\hat{y}_t) = \ell_t(W_t)$. Hence, the total loss of the algorithm can be written as

$$\hat{L}_T = \sum_{t=1}^{T} \left[ \langle s(W_t), c \rangle + \ell_t(W_t) \right].$$

Let

$$L_T(w) = T \langle s(w), c \rangle + \sum_{t=1}^{T} \ell_t(w), \qquad w \in \mathbb{R}^d,$$

be the total loss of using the weight vector $w$. Then the regret of LQDExp3 up to time $T$ on the sequence $((x_t, y_t))_{1 \le t \le T}$ can be written as

$$R_T = \max_{w \in \mathcal{W}} R_T(w),$$

where

$$R_T(w) \doteq \hat{L}_T - L_T(w), \qquad w \in \mathbb{R}^d.$$

Using the discretized weight vector set, $\mathcal{W}'$, the regret can be written as

$$R_T = \max_{w \in \mathcal{W}} R_T(w)$$

$$= \left\{ \hat{L}_T - \min_{w' \in \mathcal{W}'} L_T(w') \right\} + \left\{ \min_{w' \in \mathcal{W}'} L_T(w') - \min_{w \in \mathcal{W}} L_T(w) \right\}$$

$$= \left\{ \hat{L}_T - \min_{w' \in \mathcal{W}'} L_T(w') \right\} + \max_{w \in \mathcal{W}} \min_{w' \in \mathcal{W}'} \left\{ L_T(w') - L_T(w) \right\}. \tag{10}$$

Now, fix $w \in \mathcal{W}$. By construction, $\mathcal{W}'$ is such that for any $s \in \{0,1\}$, there exists some vector $w' \in \mathcal{W}'$ such that $s(w') = s$. Then,

$$\min_{w' \in \mathcal{W}'} \left\{ L_T(w') - L_T(w) \right\} \le \min_{w' \in \mathcal{W}' : s(w') = s(w)} \left\{ L_T(w') - L_T(w) \right\} = \min_{w' \in \mathcal{W}' : s(w') = s(w)} \sum_{t=1}^{T} \ell_t(w') - \ell_t(w).$$

Let us first deal with the second term. A simple calculation shows that $\ell_t : [-y_{\lim}, y_{\lim}] \to \mathbb{R}$, $y \mapsto (y - y_t)^2$ is $4 y_{\lim}$-Lipschitz. Hence, as long as $w' \in W'$ is such that $s(w') = s(w)$,

$$L_T(w') - L_T(w) = \sum_{t=1}^{T} \ell(\langle w', x_t \rangle, y_t) - \ell_t(\langle w, x_t \rangle, y_t) \le 4 T y_{\lim} \left( \frac{1}{T} \sum_{t=1}^{T} |\langle w - w', x_t \rangle| \right).$$

For $s \in \mathcal{G}$, define $\mathcal{W}'(s) = \{ w \in \mathcal{W}' \mid s(w) = s \}$ and $W(s) = \{ w \in \mathcal{W} \mid s(w) = s \}$. For $\alpha > 0$, let $W_\alpha(s) \subset \mathcal{W}$ be the minimal cardinality subset of $\mathcal{W}(s)$ such that $\mathrm{Lin}(\mathcal{X}, \mathcal{W}_\alpha(s))$ is an $\alpha$-cover of $\mathrm{Lin}(\mathcal{X}, \mathcal{W}(s))$ w.r.t. $d_{\mathcal{Y}}(y, y') = |y - y'|$. Choose

$$\mathcal{W}' = \cup_{s \in \mathcal{G}} \mathcal{W}_\alpha(s).$$

Then, by construction,

$$\min_{w' \in \mathcal{W}'} L_T(w') - L_T(w) \le 4 T y_{\lim} \alpha \tag{11}$$

and since this holds for any $w \in W$, we get that the same bound applies to $\max_{w \in \mathcal{W}} \min_{w' \in \mathcal{W}'} L_T(w') - L_T(w)$. Before we turn to bounding the first term of (10), let us bound the cardinality of $\mathcal{W}'$, which we will need later.

Notice that

$$|\mathcal{W}'| \le \sum_{s \in \mathcal{G}} |\mathcal{W}_\alpha(s)| \le |\mathcal{G}| \max_{s \in \mathcal{G}} |\mathcal{W}_\alpha(s)|.$$

Now, note also that thanks to the definition of $E_\mathcal{G}$, for any $s \in \mathcal{G}$, $w \in \mathcal{W}$, $\|w\|_* \le E_\mathcal{G} \cdot \|w \odot s\|_*$. Let $\mathcal{W}_\alpha$ denote a minimum cardinality $\alpha$-cover of $\mathcal{W}$. Then, it is easy to see that for any $s \in \mathcal{G}$, $\mathrm{Lin}(\mathcal{X}, \mathcal{W}_{\alpha/E_\mathcal{G}})$ is an $\alpha$-cover of $\mathrm{Lin}(\mathcal{X}, \mathcal{W}(s))$ w.r.t. $d_{\mathcal{Y}}(y, y') = |y - y'|$. Hence, by the minimum cardinality property of $\mathcal{W}_\alpha(s)$, we have $|\mathcal{W}_\alpha(s)| \le |\mathcal{W}_{\alpha/E_\mathcal{G}}|$ and, by Lemma 2.1, we get that $\ln^+ |\mathcal{W}_\alpha(s)| \le d \ln(1 + 2 E_\mathcal{G} y_{\lim}/\alpha)$. Hence,

$$\ln |\mathcal{W}'| \le \ln(|\mathcal{G}|) + d \ln(1 + 2 E_\mathcal{G} y_{\lim}/\alpha). \tag{12}$$

Let us now turn to bounding the expectation of the first term of (10). We have,

$$\mathbb{E}\left[\hat{L}_T - \min_{w \in \mathcal{W}'} L_T(w)\right] = \max_{w \in \mathcal{W}'} \mathbb{E}\left[\hat{L}_T - L_T(w)\right],$$

where we have exploited that $L_T(w)$ is deterministic. Therefore, it suffices to bound $\mathbb{E}\left[\hat{L}_T - L_T(w)\right]$ for any fixed $w \in \mathcal{W}'$. Thus, fix $w \in \mathcal{W}'$.

By the construction of $\tilde{\ell}_t$, $\mathbb{E}\left[\tilde{\ell}_t(w)|p_t\right] = \ell_t(w)$ holds for any $w \in \mathbb{R}^d$. Hence, $\mathbb{E}\left[L_T(w)\right] = \mathbb{E}\left[\sum_{t=1}^T \tilde{\ell}_t(w)\right]$. Furthermore, $\mathbb{E}\left[\ell_t(W_t)|p_t\right] = \sum_{w \in \mathcal{W}'} p_t(w)\ell_t(w) = \sum_{w \in \mathcal{W}'} p_t(w)\mathbb{E}\left[\tilde{\ell}_t(w)|p_t\right] = \mathbb{E}\left[\sum_{w \in \mathcal{W}'} p_t(w)\tilde{\ell}_t(w)|p_t\right]$.

Introduce $\hat{\tilde{\ell}}_t(w) = \tilde{\ell}_t(w) + \langle \mathrm{s}(w), c \rangle$, Then, we see that it suffices to bound

$$\mathbb{E}\left[\hat{L}_T - L_T(w)\right] = \mathbb{E}\left[\sum_{t=1}^T \sum_{w' \in W'} p_t(w')\hat{\tilde{\ell}}_t(w') - \sum_{t=1}^T \hat{\tilde{\ell}}_t(w)\right].$$

Now, by Lemma A.2.1, under the assumption that $0 < \gamma \leq 1, 0 < \eta$ are such that for any $w' \in \mathcal{W}'$, $1 \leq t \leq T$ the inequality

$$\eta\hat{\tilde{\ell}}_t(w') \geq -1 \tag{13}$$

holds, we have

$$\sum_{t=1}^T \sum_{w' \in W'} p_t(w')\hat{\tilde{\ell}}_t(w') - \sum_{t=1}^T \hat{\tilde{\ell}}_t(w)$$

$$\leq \frac{\ln|\mathcal{W}'|}{\eta} + \eta \sum_{t=1}^T \sum_{w' \in \mathcal{W}'} p_t(w')\hat{\tilde{\ell}}_t^2(w') + \gamma \sum_{t=1}^T \sum_{w' \in \mathcal{W}'} \mu(w')(\hat{\tilde{\ell}}_t(w') - \hat{\tilde{\ell}}_t(w)).$$

Let us assume for a moment that $\eta, \gamma$ can be chosen to satisfy the above quoted conditions – we shall return to the choice of these parameters soon. Taking expectations of both sides of the last inequality, we get

$$\mathbb{E}\left[\hat{L}_T - L_T(w)\right] \leq \frac{\ln|\mathcal{W}'|}{\eta} + \eta \sum_{t=1}^T \sum_{w' \in \mathcal{W}'} \mathbb{E}\left[p_t(w')\hat{\tilde{\ell}}_t^2(w')\right] + \gamma \sum_{t=1}^T \sum_{w' \in \mathcal{W}'} \mu(w')(\ell_t(w') + \langle \mathrm{s}(w'), c \rangle),$$

where we have used that $\mathbb{E}\left[\hat{\tilde{\ell}}_t(w)\right] = \ell_t(w) + \langle \mathrm{s}(w), c \rangle \geq 0$. Thus, we see that it remains to bound $\mathbb{E}\left[p_t(w')\hat{\tilde{\ell}}_t^2(w')\right]$, which is done in the following lemma.

**Lemma A.2.2.** *Let $\mathcal{W}'$, $\tilde{\ell}_t$, $p_t$ be as in LQDEXP3. Then, there exist a constant $C > 0$ such that the following equation holds:*

$$\sum_{w \in \mathcal{W}'} p(w)\mathbb{E}\left[\hat{\tilde{\ell}}^2(w) \,|\, p\right] \leq (4y_{\mathrm{lim}}^2 + \|c\|_1)(W_\infty^2 X_1^2 + 2y_{\mathrm{lim}}W_\infty X_1 + y_{\mathrm{lim}}^2 + \|c\|_1).$$

*Proof.* By the tower rule, we have

$$\mathbb{E}\left[\sum_{w \in \mathcal{W}'} p_t(w)\hat{\tilde{\ell}}_t^2(w)\right] = \mathbb{E}\left[\sum_{w \in \mathcal{W}'} p_t(w)\mathbb{E}\left[\hat{\tilde{\ell}}_t^2(w) \,|\, p_t\right]\right]$$

Therefore, it suffices to bound

$$\sum_{w \in \mathcal{W}'} p_t(w)\mathbb{E}\left[\hat{\tilde{\ell}}_t^2(w) \,|\, p_t\right].$$

For simplifying the presentation, since $t$ is fixed, from now on we will remove the subindex $t$ from the quantities involved and write $\hat{\tilde{\ell}}$ instead of $\hat{\tilde{\ell}}_t$, $p$ instead of $p_t$, etc.

The plan of the proof is as follows: We construct a deterministic upper bound $h(w)$ on $|\hat{\tilde{\ell}}(w)|$ and an upper bound $B$ on $\sum_{w\in\mathcal{W}'} p(w)h(w)$. Then, we provide an upper bound $B'$ on $\mathbb{E}\left[\hat{\tilde{\ell}}(w)|p\right]$ so that

$$\sum_{w\in\mathcal{W}'} p(w)\mathbb{E}\left[\hat{\tilde{\ell}}^2(w)\,\Big|\,p\right] \leq \sum_{w\in\mathcal{W}'} p(w)h(w)\mathbb{E}\left[\hat{\tilde{\ell}}(w)\,\Big|\,p\right] \leq B'\sum_{w\in\mathcal{W}'} p(w)h(w) \leq BB'.$$

Before providing these bounds, let's review some basic relations. Remember that $W_\infty = \sup_{w\in\mathcal{W}}\|w\|_\infty$ and $X_1 = \sup_{x\in\mathcal{X}}\|x\|_1$. Further, note that for any $1 \leq j, j' \leq d$, we have

$$\mathbb{E}\left[\mathbb{1}_{\{j\in s(W)\}}\,\Big|\,p\right] = \sum_{w\in\mathcal{W}':j\in s(w)} p(w) = \sum_{w\in\mathcal{W}'} \mathbb{1}_{\{j\in s(w)\}}p(w) = q(j), \tag{14}$$

$$\mathbb{E}\left[\mathbb{1}_{\{j,j'\in s(W)\}}\,\Big|\,p\right] = \sum_{w\in\mathcal{W}':j,j'\in s(w)} p(w) = \sum_{w\in\mathcal{W}'} \mathbb{1}_{\{j,j'\in s(w)\}}p(w) = q(j,j'). \tag{15}$$

As to the upper bound $h(w)$ on $|\hat{\tilde{\ell}}(w)|$, we start with

$$|\hat{\tilde{\ell}}(w)| \leq |w^\top \tilde{X}w| + 2|y|\,|w^\top \tilde{x}| + |y|^2 + \|c\|_1. \tag{16}$$

Now, $|y| \leq y_{\lim}$ and

$$|w^\top \tilde{x}| \leq W_\infty \sum_{j=1}^{d} \mathbb{1}_{\{j\in s(w)\}}\frac{|x_j|}{q(j)} \doteq g(w,x),$$

$$|w^\top \tilde{X}w| \leq W_\infty^2 \sum_{j,j'} \mathbb{1}_{\{j,j'\in s(w)\}}\frac{|x_jx_{j'}|}{q(j,j')} \doteq G(w,x).$$

Hence,

$$|\hat{\tilde{\ell}}(w)| \leq G(w,x) + 2y_{\lim}g(w,x) + y_{\lim}^2 + \|c\|_1 \doteq h(w)$$

which is indeed a deterministic upper bound on $|\hat{\tilde{\ell}}(w)|$. To bound $\sum_{w\in\mathcal{W}'} p(w)h(w)$, it remains to upper bound $\sum_{w\in\mathcal{W}'} p(w)g(w,x)$ and $\sum_{w\in\mathcal{W}'} p(w)G(w,x)$. To upper bound these, we move the sum over the weights $w$ inside the other sums in the definitions of $g$ and $G$ to get:

$$\sum_{w\in\mathcal{W}'} p(w)g(w,x) = W_\infty \sum_{j=1}^{d} \frac{|x_j|}{q(j)} \sum_{w\in\mathcal{W}'} p(w)\mathbb{1}_{\{j\in s(w)\}} = W_\infty X_1, \qquad \text{(by (14) and } \|x\|_1 \leq X_1\text{)}$$

$$\sum_{w\in\mathcal{W}'} p(w)G(w,x) = W_\infty^2 \sum_{j,j'} \frac{|x_jx_{j'}|}{q(j,j')} \sum_{w\in\mathcal{W}'} p(w)\mathbb{1}_{\{j,j'\in s(w)\}} = W_\infty^2 \sum_{j,j'} |x_jx_{j'}| \qquad \text{(by (15))}$$

$$= W_\infty^2\|x\|_1^2 \leq W_\infty^2 X_1^2.$$

Hence,

$$\sum_{w\in\mathcal{W}'} p(w)h(w) \leq W_\infty^2 X_1^2 + 2y_{\lim}W_\infty X_1 + y_{\lim}^2 + \|c\|_1.$$

Let us now turn to bounding $\mathbb{E}\left[|\hat{\tilde{\ell}}(w)|\,\Big|\,p\right]$. From (16), it is clear that it suffices to upper bound $\mathbb{E}\left[|w^\top \tilde{X}w|\,|\,p\right]$ and $\mathbb{E}\left[|w^\top \tilde{x}|\,|\,p\right]$. From (14) and (15), the definitions of $\tilde{x}$ and $\tilde{X}$ and because by assumption $\||w\|\|_*\||x\|\| \leq w_{\lim}x_{\lim} = y_{\lim}$, we obtain

$$\mathbb{E}\left[|w^\top \tilde{x}|\,|\,p\right] = \sum_j |w_jx_j| \leq y_{\lim} \quad \text{and}$$

$$\mathbb{E}\left[|w^\top \tilde{X}w|\,|\,p\right] = \sum_{j,j'} |w_jw_{j'}x_jx_{j'}| = \left(\sum_j |w_jx_j|\right)^2 \leq y_{\lim}^2.$$

Thus,

$$\mathbb{E}\left[|\hat{\tilde{\ell}}(w)| \, \Big| \, p\right] \le \mathbb{E}\left[|w^\top \tilde{X} w| + 2y_{\text{lim}}|w^\top \tilde{x}| + y_{\text{lim}}^2 + \|c\|_1 \, \Big| \, p\right] \le 4y_{\text{lim}}^2 + \|c\|_1.$$

Putting together all the bounds, we get

$$\sum_{w \in \mathcal{W}'} p(w)\mathbb{E}\left[\hat{\tilde{\ell}}^2(w) \,|\, p\right] \le (4y_{\text{lim}}^2 + \|c\|_1)(W_\infty^2 X_1^2 + 2y_{\text{lim}}W_\infty X_1 + y_{\text{lim}}^2 + \|c\|_1).$$

$\square$

It remains to bound $\sum_{w' \in \mathcal{W}'} \mu(w')\left(\ell_t(w') + \langle \text{s}(w'), c \rangle\right)$. Because of the bounds on weight vectors in $\mathcal{W}'$ and $((x_t, y_t))_{(1 \le t \le T)}$, we know that $\ell_t(w') + \langle \text{s}(w'), c \rangle \le 4y_{\text{lim}}^2 + \|c\|_1$. Combining the inequalities obtained so far, we get

$$\mathbb{E}\left[\hat{L}_T - L_T(w)\right] \le \frac{\ln|\mathcal{W}'|}{\eta} + \eta(4y_{\text{lim}}^2 + \|c\|_1)(W_\infty^2 X_1^2 + 2y_{\text{lim}}W_\infty X_1 + y_{\text{lim}}^2 + \|c\|_1) \qquad (17)$$
$$+ \gamma T(4y_{\text{lim}}^2 + \|c\|_1).$$

Thus, it remains to select $\eta, \gamma$ such that the earlier imposed conditions, amongst them (13), hold and the above bound on the expected regret is minimized. To ensure $\eta \hat{\tilde{\ell}}_t(w) \ge -1$, we start with a lower bound on $\tilde{\ell}_t(w)$:

$$\tilde{\ell}_t(w) = w^\top \tilde{X}_t \, w - 2\, w^\top \tilde{x}_t \, y_t + y_t^2$$
$$\ge w^\top \tilde{X}_t \, w - 2\, w^\top \tilde{x}_t \, y_t$$
$$= \sum_{i,j=1}^d w_i w_j (\tilde{X}_t)_{i,j} - 2y_t \sum_{j=1}^d w_j \tilde{x}_{t,j}$$
$$\ge -\frac{\sum_{i,j} \mathbb{1}_{\{i \in \text{s}(w)\}} \mathbb{1}_{\{j \in \text{s}(w)\}} |x_{t,i} x_{t,j} w_i w_j|}{\gamma} - 2y_{\text{lim}} \frac{\sum_i \mathbb{1}_{\{i \in \text{s}(w)\}} |x_{t,i} w_i|}{\gamma}$$
$$\ge -\frac{\|w\|_*^2 \|x_t\|^2}{\gamma} - 2y_{\text{lim}} \frac{\|w\|_* \|x_t\|}{\gamma}$$
$$\ge -\frac{3y_{\text{lim}}^2}{\gamma}.$$

Thus, as long as $3\eta y_{\text{lim}}^2 \le \gamma$, it follows that (13) holds. To minimize (17), we choose

$$\gamma = 3\eta y_{\text{lim}}^2 \qquad (18)$$

to get

$$\mathbb{E}\left[\hat{L}_T - L_T(w)\right] \le \frac{\ln|\mathcal{W}'|}{\eta} + \eta(4y_{\text{lim}}^2 + \|c\|_1)(W_\infty^2 X_1^2 + 2y_{\text{lim}}W_\infty X_1 + 4y_{\text{lim}}^2 + \|c\|_1).$$

Using $\eta = \sqrt{\frac{\ln|\mathcal{W}'|}{(4y_{\text{lim}}^2 + \|c\|_1)(W_\infty^2 X_1^2 + 2y_{\text{lim}}W_\infty X_1 + 4y_{\text{lim}}^2 + \|c\|_1)}}$, we get

$$\mathbb{E}\left[\hat{L}_T\right] - L_T(w) \le 2\sqrt{T(4y_{\text{lim}}^2 + \|c\|_1)(W_\infty^2 X_1^2 + 2y_{\text{lim}}W_\infty X_1 + 4y_{\text{lim}}^2 + \|c\|_1)\ln|\mathcal{W}'|}.$$

Noting that here $w \in \mathcal{W}'$ was arbitrary, together with the regret decomposition (10), the bound (11) on the regret arising from discretization the bound (12) on $\ln|\mathcal{W}'|$ and that $\ln|\mathcal{G}| \le d\ln 2$ give

$$\mathbb{E}[R_T] \le 2\sqrt{T(4y_{\text{lim}}^2 + \|c\|_1)(W_\infty^2 X_1^2 + 2y_{\text{lim}}W_\infty X_1 + 4y_{\text{lim}}^2 + \|c\|_1)\ln|\mathcal{W}'|} + 4\,T\,y_{\text{lim}}\alpha$$

$$\le 2\sqrt{Td(4y_{\text{lim}}^2 + \|c\|_1)(W_\infty^2 X_1^2 + 2y_{\text{lim}}W_\infty X_1 + 4y_{\text{lim}}^2 + \|c\|_1)\ln(2 + 4E_\mathcal{G}y_{\text{lim}}/\alpha)} + 4\,T\,y_{\text{lim}}\alpha.$$

Choosing $\alpha = y_{\text{lim}}T^{-1/2}$, we get the bound

$$\mathbb{E}[R_T] \le C\sqrt{Td(4y_{\text{lim}}^2 + \|c\|_1)(W_\infty^2 X_1^2 + 2y_{\text{lim}}W_\infty X_1 + 4y_{\text{lim}}^2 + \|c\|_1)\ln(E_\mathcal{G}T)}. \qquad (19)$$

for some constant $C > 0$. $\qquad \square$

### A.2.2 Lower Bound

**Theorem 2.5.** *Let $d > 0$, and consider the online free label probing problem with linear predictors, where $\mathcal{W} = \{w \in \mathbb{R}^d \mid \|w\|_1 \le w_{\lim}\}$ and $\mathcal{X} = \{x \in \mathbb{R}^d \mid \|x\|_\infty \le 1\}$. Assume, for all $t \ge 1$, that the loss functions are of the form $\ell_t(w) = (w^\top x_t - y_t)^2 + \langle \mathrm{s}(w), c \rangle$, where $|y_t| \le 1$ and $c = 1/2 \times \mathbf{1} \in \mathbb{R}^d$. Then, for any prediction algorithm and for any $T \ge \frac{4d}{8\ln(4/3)}$, there exists a sequence $((x_t, y_t))_{1 \le t \le T} \in (\mathcal{X} \times [-1, 1])^T$ such that the regret of the algorithm can be bounded from below as*

$$\mathbb{E}[R_T] \ge \frac{\sqrt{2} - 1}{\sqrt{32 \ln(4/3)}} \sqrt{Td} \,.$$

*Proof.* The idea of the proof is similar to Mannor and Shamir (2011, Theorem 4). We will solve the problem of Multi-Armed Bandits with $d$ arms using an algorithm that can solve free-label probing with examples having $d$ features. We will use the lower bound proved in Cesa-Bianchi and Lugosi (2006, Theorem 6.11) for Multi-Armed Bandit game. They showed a method of choosing the losses and proved that there exist a universal constant $C_{MAB}$ such that no algorithm can achieve a better regret than $C_{MAB}\sqrt{Td}$ in $T$ rounds using $d$ arms. In their method adversary chooses one of the arms beforehand and assign a random Bernoulli loss with parameter $1/2 + \varepsilon$ to that arm and a random Bernoulli loss with parameter $1/2$ to all other arms at each round. Then they proved that by choosing $\varepsilon = \sqrt{(1/(8\ln(4/3))d/T}$, no algorithm can achieve better expected regret bound than $C_{MAB}\sqrt{Td}$ in $T$ rounds. Note that they use the fact that losses are in range $[0, 1]$. Without loss of generality we can add $1/2$ to all the losses and assume that the losses are now in range $[1/2, 1 + 1/2]$ and their result still hold.

Now we explain how we can solve that problem using an algorithm that solves free label probing game. More formally we will use the following lemma.

**Lemma A.2.3.** *Give any learner $\mathcal{A}$ for an online free-label probing game there exist a learner $\mathcal{A}'$ for Multi-Armed Bandit problem with the adversaries proposed at Cesa-Bianchi and Lugosi (2006, Theorem 6.11) and an adversary for online free-label probing game such that*

$$\mathbb{E}\left[R_{\mathcal{A}'}(T, MAB)\right] - 2d\sqrt{(1/(8\ln(4/3))} \le \mathbb{E}\left[R_{\mathcal{A}}(T, OFLP)\right] \,,$$

*holds where $R_{\mathcal{A}'}(T, MAB)$ is the regret of the learner $\mathcal{A}'$ in the Multi-Armed Bandit problem with the defined adversary and $R_{\mathcal{A}}(T, OFLP)$ is the regret of the learner $\mathcal{A}$ in the online free-label probing game.*

*Proof.* We define the adversary in the online free label probing game. The adversary chooses $y_t = 1$ for all the rounds. Note the the challenge is finding a weight vector to predict the label and not only predicting the label. Consider the weight vector $e_i$ which is a zero weight vector with a single one in its $i$th element for all $1 \le i \le d$. The adversary then chooses one of the components $v$, in advance and sets $x_{t,i}$ to be a Bernoulli random variable with parameter one for every $i \ne v$ and sets $x_{t,v}$ to be a Bernoulli random variable with parameter $1/2 + \varepsilon$. Note that this component $v$ is the same arm as the adversary in multi-arm bandit chooses. Now we know that for each $e_i$ the loss will be the cost of observing $i$th feature which is $1/2$ and a prediction error which is a Bernoulli random variable based on the assignments to the features. So you can easily see a correspondence between $e_i$ and $i$th arm in multi-armed bandit problem with the adversary defined in Cesa-Bianchi and Lugosi (2006, Theorem 6.11).

Let $R_{\mathcal{A}}(T, OFLP)$ denote the regret of the learner $\mathcal{A}$ in this online free-label probing. We know that if we make the set of competitors smaller, the regret can not be increased. Note that we does not change the set of actions that algorithm $\mathcal{A}$ can take. Let $R^*_{\mathcal{A}}(T, OFLP)$ denote the regret of the learner $\mathcal{A}$ in this online free-label probing when it competes only against $e_i$ weight vectors for all $1 \le i \le d$. Since we make the set of competitors smaller we have

$$R^*_{\mathcal{A}}(T, OFLP) \le R_{\mathcal{A}}(T, OFLP) \,. \tag{20}$$

Now consider the learner $\mathcal{A}$ that solves this online free-label probing game. We will construct another algorithm $\mathcal{A}'$ such that solves the multi-armed bandits problem. Let $I_t$ denote the chosen arm by $\mathcal{A}'$ and $\ell_{t,i}$ denote the loss of arm $i$ at round $t \ge 1$. Here are the different situations.

When $\mathcal{A}$ chooses $w_t = \mathbf{0} \in \mathbb{R}^d$ at round $t$, $\mathcal{A}'$ chooses one of the arms randomly in multi-armed bandit problem. By this choice, $\mathcal{A}$ does not observe any feature and predict zero for the label. Here is the expected regret at these types of rounds for $\mathcal{A}$.

$$\mathbb{E}\left[\ell_t(\mathbf{0}) - \ell_t(e_v)\right] = 1 - (1/2 + \mathbb{E}\left[(e_v^\top x_t - y_t)^2\right] = 1 - (1/2 + 1/2 - \varepsilon) = \varepsilon \,.$$

On the other hand, the expected regret of $\mathcal{A}'$ in the game of multi-armed bandits at each round is bounded by $\varepsilon$. By this we know that in the rounds that $\mathcal{A}$ chooses $w_t = \mathbf{0} \in \mathbb{R}^d$ we get

$$\mathbb{E}\left[\ell_{t,I_t} - \ell_{t,v}\right] = \mathbb{E}\left[\ell_t(e_{I_t}) - \ell_t(e_v)\right] \leq \varepsilon = \mathbb{E}\left[\ell_t(w_t) - \ell_t(e_t)\right] \qquad t \geq 1 \,, \tag{21}$$

which means the regret of $\mathcal{A}'$ is not going to be increased more that regret of $\mathcal{A}$ in such rounds.

When $\mathcal{A}$ chooses a weight vector $w_t \neq \mathbf{0}$, $\mathcal{A}'$ chooses all arms $i$ in the bandit game whose corresponding $i$th component of $w_t$ is not zero in the free-label probing game in the consecutive rounds and after finding all required component values of $x$, it gives it to $\mathcal{A}$ as the feedback for calculating the loss. Note that the chosen weight vector by $\mathcal{A}$ requires either one feature or more than one feature. As a result $\mathcal{A}'$ plays the bandit games for $T'$ rounds while $\mathcal{A}$ plays the online free-label probing game for $T$ rounds. If it $w_t$ needs only one feature due to the way the choice of $x_{t,i}$, the minimizer of expected loss is exactly $e_i$. Because if the $i$th component of $w_t$ was $\alpha$ instead of one we get

$$\mathbb{E}\left[(w_t^\top x_t - y_t)^2\right] = \mathbb{E}\left[(\alpha x_{t,i} - 1)^2\right] = \mathbb{P}\left[x_{t,i} = 0\right] \times 1 + \mathbb{P}\left[x_{t,i} = 1\right] \times (1 - \alpha)^2 \,.$$

which achieves its minimum for $\alpha = 1$. So we get Eq.(21) for these types of rounds as well. Now if $w_t$ has more than one non-zero components as we said $\mathcal{A}'$ plays more rounds. At these extra rounds the expected regret of $\mathcal{A}'$ will be increased by at most $\varepsilon$. However $\mathcal{A}$ is also paying for those extra features that it needed. Since the cost of each feature is $1/2$ as well assuming that $\varepsilon \leq 1/2$, we can conclude that the regret of $\mathcal{A}$ for all these extra rounds is still less than or equal the regret of $\mathcal{A}$ on the rounds that it chooses $w_t$. Let $T'$ denote the random number of rounds that $\mathcal{A}'$ is playing the bandits game. We know that this number is bounded by $dT$ since at each round $\mathcal{A}$ can choose at most all the features. Putting the above results together with Eq.(21), we get

$$\mathbb{E}\left[R_{\mathcal{A}'}(T', MAB)\right] \leq \mathbb{E}\left[R_{\mathcal{A}}^*(T, OFLP)\right] \,.$$

Because the expected regret is increasing in the number of rounds we can use $\mathbb{E}\left[R_{\mathcal{A}'}(T, MAB)\right] \leq \mathbb{E}\left[R_{\mathcal{A}'}(T', MAB)\right]$ and also Eq.(20) to get

$$\mathbb{E}\left[R_{\mathcal{A}'}(T, MAB)\right] \leq \mathbb{E}\left[R_{\mathcal{A}}(T, OFLP)\right] \,.$$

Using the value of $\varepsilon$ that Cesa-Bianchi and Lugosi (2006, Theorem 6.11) uses we get the lemma statement. Also $T >= \frac{4d}{8\ln(4/3)}$ in the lemma statement guarantees that $\varepsilon \leq 1/2$ which was needed in the middle of the proof. $\qquad \square$

Using this lemma and also knowing that

$$\mathbb{E}\left[R_{\mathcal{A}'}(T, MAB)\right] \geq \sqrt{dT} \frac{\sqrt{2} - 1}{\sqrt{32\ln(4/3)}}$$

based on the result of Cesa-Bianchi and Lugosi (2006, Theorem 6.11), we can derive

$$\frac{\sqrt{2} - 1}{\sqrt{32\ln(4/3)}} \sqrt{dT} \leq \mathbb{E}\left[R_{\mathcal{A}}(T, OFLP)\right] \,.$$

$\qquad \square$

## A.3 Non-Free-Label Probing

### A.3.1 Revealing Action Algorithm for Non-Free-Label Probing

**Algorithm 3** Revealing action algorithm for non-free-label online probing

---

**Parameters:** Real numbers $0 \leq \eta, \gamma \leq 1$, Set of experts $\mathcal{F}$.
**Initialization:** $u_1(f) = 1$ $(f \in \mathcal{F})$.
**for** $t = 1$ **to** $T$ **do**
    Draw $F_t \in \mathcal{F}$ from the probability mass function

$$p_t(f) = \frac{u_t(f)}{\sum_{f \in \mathcal{F}} u_t(f)}, \qquad f \in \mathcal{F}.$$

    Draw a Bernoulli random variable $Z_t$ such that $\mathbb{P}\left[Z_t = 1\right] = \gamma$.
    **if** $Z_t = 0$ **then**
        $S_t = (\mathrm{s}(F_t), 0)$ (*i.e.,* $s_{t,d+1} = 0$).
        Obtain the features values, $(x_{t,i})_{i \in \mathrm{s}(F_t)}$.
        Predict $\hat{y}_t = F_t(x_t)$.
    **else**
        $S_t = \mathbf{1} \in \mathbb{R}^{d+1}$ (*i.e.,* all $d+1$ components are one).
        Observe all the features of $x_t$.
        Predict $\hat{y}_t = F_t(x_t)$.
        Receive the true label $y_t$.
    **end if**
    **for each** $f \in \mathcal{F}$ **do**
        $\tilde{\ell}_t(f) = \mathbb{1}_{\{Z_t=1\}} \frac{\langle \mathrm{s}(f), c_{1:d} \rangle + \ell_t(\hat{y}_t)}{\gamma}$.
        $u_{t+1}(f) = u_t(f) \exp(-\eta \tilde{\ell}_t(f))$.
    **end for**
**end for**

---

### A.3.2  Upper Bound

**Lemma 3.1.** *Given any non-free-label online probing with finitely many experts, Algorithm 3 with appropriately set parameters achieves*

$$\mathbb{E}[R_T] \leq C \max\left(T^{2/3}(\ell_{\max}^2 \|c\|_1 \ln |\mathcal{F}|)^{1/3}, \ell_{\max}\sqrt{T \ln |\mathcal{F}|}\right)$$

*for some constant $C > 0$.*

*Proof.* The regret of the algorithm is decomposed into two additive terms: (i) The extra loss suffered in exploration rounds. The cumulative expectation of this extra loss can be upper bounded by $T\gamma\|c\|_1$. (ii)The regret of the algorithm compared to each expert, excluding rounds that request the label and extra features. To upper bound this term, we follow the classical "exponential weights" proof (see *e.g.,* Cesa-Bianchi et al. (2006)).

First we make the trivial observation that for every time step $t$ and $f \in \mathcal{F}, \mathbb{E}[\tilde{\ell}_t(f)] = \langle \mathrm{s}(f), c_{1:d} \rangle + \ell_t(f(s \odot x_t))$. That is, $\tilde{\ell}_t(f)$ is an unbiased estimate of the true loss of function $f$. Let $U_t = \sum_{f \in \mathcal{F}} u_t(f)$. Now we continue with lower and upper bounding the term $U_T$:

$$U_T \geq \sum_{f \in \mathcal{F}} u_T(f) \geq u_T(f^*) = \exp\left(-\eta \sum_{t=1}^{T} \tilde{\ell}_t(f^*)\right),$$

where $f^*$ is and arbitrary expert in $\mathcal{F}$. For the upper bound we write

$$
\begin{aligned}
\frac{U_t}{U_{t-1}} &= \sum_{f \in \mathcal{F}} \frac{u_{t-1}(f) \exp(-\eta \tilde{\ell}_t(f))}{U_{t-1}} \\
&= \sum_{f \in \mathcal{F}} p_t(f)(1 - \eta \tilde{\ell}_t(f) + \eta^2 \tilde{\ell}_t^2(f)) \qquad\qquad (22) \\
&= 1 - \eta \sum_{f \in \mathcal{F}} p_t(f) \tilde{\ell}_t(f) + \eta^2 \sum_{f \in \mathcal{F}} p_t(f) \tilde{\ell}_t^2(f) \\
&\leq \exp\left(-\eta \sum_{f \in \mathcal{F}} p_t(f) \tilde{\ell}_t(f) + \eta^2 \sum_{f \in \mathcal{F}} p_t(f) \tilde{\ell}_t^2(f)\right), \qquad\qquad (23)
\end{aligned}
$$

where in (22) we used that $u_{t-1}(f)/U_{t-1} = p_t(f)$ and the inequality $e^x \leq 1 + x + x^2$ if $x \leq 1$, and in (23) we used that $e^x \geq 1 + x$. Multiplying the above inequality for $t = 1, \ldots, T$ and also $U_1$ we get

$$
U_T \leq |\mathcal{F}| \exp\left(-\eta \sum_{t=1}^{T} \sum_{f \in \mathcal{F}} p_t(f) \tilde{\ell}_t(f) + \eta^2 \sum_{t=1}^{T} \sum_{(s,f(.)) \in \mathcal{F}} p_t(f) \tilde{\ell}_t^2(f)\right).
$$

We now merge the lower and upper bounds and take logarithm of both sides:

$$
-\eta \sum_{t=1}^{T} \tilde{\ell}_t(f^*) - \ln |\mathcal{F}| \leq -\eta \sum_{t=1}^{T} \sum_{f \in \mathcal{F}} p_t(f) \tilde{\ell}_t(f) + \eta^2 \sum_{t=1}^{T} \sum_{f \in \mathcal{F}} p_t(f) \tilde{\ell}_t^2(f).
$$

Rearranging gives

$$
\sum_{t=1}^{T} \sum_{f \in \mathcal{F}} p_t(f) \tilde{\ell}_t(f) - \sum_{t=1}^{T} \tilde{\ell}_t(f^*) \leq \eta \sum_{t=1}^{T} \sum_{f \in \mathcal{F}} p_t(f) \tilde{\ell}_t^2(f) + \frac{\ln |\mathcal{F}|}{\eta}.
$$

After taking expectation of both sides, the first term on the left hand side is the expected cumulative loss of the algorithm excluding the extra loss suffered in exploration rounds, while the second term is the expected cumulative loss of the any arbitrary expert $f$. The first term on the right hand side can be upper bounded as

$$
\begin{aligned}
\eta \sum_{t=1}^{T} \sum_{f \in \mathcal{F}} \mathbb{E}[p_t(f) \tilde{\ell}_t^2(f)] &\leq \eta \sum_{t=1}^{T} \sum_{f \in \mathcal{F}} \mathbb{E}[p_t(f) \tilde{\ell}_t(f)] \frac{\ell_{\max}}{\gamma} \\
&\leq \frac{\eta \ell_{\max}^2 T}{\gamma},
\end{aligned}
$$

where $\ell_{\max}$ is the maximum loss an action can suffer, ignoring the label cost $c_{d+1}$.

Adding up the two terms of the expected regret, we get

$$
\mathbb{E}[R_T] \leq T\gamma \|c\|_1 + \frac{\eta \ell_{\max}^2 T}{\gamma} + \frac{\ln |\mathcal{F}|}{\eta}.
$$

For setting the parameters optimally, we consider two cases.

(1) If $\|c\|_1 \geq \sqrt{\frac{\ln |\mathcal{F}|}{2T}} \ell_{\max}$, then we set

$$
\eta = (\ln |\mathcal{F}|)^{2/3} T^{-2/3} (4 \ell_{\max}^2 \|c\|_1)^{-1/3} \qquad\qquad \gamma = \sqrt{\frac{\eta \ell_{\max}^2}{\|c\|_1}}
$$

to get

$$
\mathbb{E}[R_T] \leq C_1 T^{2/3} (\ell_{\max}^2 \|c\|_1 \ln |\mathcal{F}|)^{1/3}
$$

for some constant $C_1 > 0$. The condition on $\|c\|_1$ is needed for $\gamma$ to be a probability. On the other hand,

(2) if $\|c\|_1 < \sqrt{\frac{\ln |\mathcal{F}|}{2T}} \ell_{\max}$, then we set

$$\eta = \sqrt{\frac{\ln |\mathcal{F}|}{T \ell_{\max}^2}} \qquad\qquad \gamma = 1$$

to get

$$\mathbb{E}[R_T] \leq C_2 \ell_{\max} \sqrt{T \ln |\mathcal{F}|}$$

for some constant $C_2$.

Combining the two bounds gives the result of the lemma. $\qquad\square$

### A.3.3 Lower Bound

In this section we prove the lower bound on the regret of the non-free-label probing game, stated in Section 3. The proof follows a standard lower bounding technique using a randomized construction for the loss functions. As such, we omit the proofs of two lemmas used in the derivation; the interested reader is referred to (Bartók, 2012) for these proofs.

**Theorem 3.2.** *There exists a constant $C$ such that, for any non-free-label probing with linear predictors, quadratic loss, and $c_j > (1/d) \sum_{i=1}^{d} c_i - 1/2d$ for every $j = 1, \ldots, d$, the expected regret of any algorithm can be lower bounded by*

$$\mathbb{E}[R_T] \geq C(c_{d+1}d)^{1/3}T^{2/3} .$$

*Proof.* We construct a set of opponent strategies and show that the expected regret of any algorithm is high against at least one of them. The features $x_{t,i}$ for $t = 1, \ldots, T$ and $i = 1, \ldots, d$ are generated by the iid random variables $X_{t,i}$ whose distribution is Bernoulli with parameter 0.5. Let $Z_t \in \{1, \ldots, d\}$ be random variables whose distribution will be specified later. The labels $y_t$ are generated by the random variable defined as $Y_t = X_{t,Z_t}$.

To construct the distribution of $Z_t$ we introduce the following notation. For every $i = 1, \ldots, d$, let

$$a_i = \frac{1}{d} + 2c_i - \frac{2}{d} \sum_{j=1}^{d} c_j .$$

The assumptions on $c$ ensures that $a_i > 0$ for every $i = 1, \ldots, d$. For opponent strategy $k$, let the distribution of $Z_t$ defined as

$$\mathbb{P}_k [Z_t = i] = \begin{cases} a_i - \varepsilon, & i \neq k; \\ a_i + (d-1)\varepsilon, & \text{i=k} , \end{cases}$$

with some $\varepsilon > 0$ to be defined later.

**Lemma A.3.1.** (Bartók 2012, Lemma 25) *Let $e_k$ denote the $k^{\text{th}}$ basis vector of dimension d. Against opponent strategy k, the instantaneous expected regret for any action such that $(s, s_\ell) \neq (e_k, 0)$ is at least $\frac{d\varepsilon}{2}$.*

For $i = 1, \ldots, d$, let $N_i$ denote the number of times the player's action is $(e_i, w, s_{d+1})$. Similarly, let $N_L$ denote the number of times the player requests the label. Now it is easy to see that the expected regret under opponent strategy $k$ can be lower bounded by

$$\mathbb{E}_k[R_T] \geq (T - \mathbb{E}_k[N_k]) \frac{d\varepsilon}{2} + c_{d+1}\mathbb{E}_k[N_L] .$$

The rest of the proof is devoted to show that for any algorithm, the average of the above value, $1/d \sum_{i=1}^{d} \mathbb{E}_i[R_T]$ can be lower bounded. We only show this for deterministic algorithms. The statement follows for randomizing algorithms with the help of a simple argument, see *e.g.,* Cesa-Bianchi and Lugosi (2006, Theorem 6.11).

A deterministic algorithm is defined as a sequence of functions $A_t(\cdot)$, where the argument of $A_t$ is a sequence of observations up to time step $t - 1$ and the value is the action taken at time step $t$. We

denote the observation at time step $t$ by $h_t \in \{0, 1, *\}^{d+1}$, where $h_{t,i} = x_{t,i}$ if $s_{t,i} = 1$ and $h_{t,i} = *$ if $s_{t,i} = 0$ for all $1 \le i \le d$. Similarly, $h_{t,d+1} = y_t$ if $s_{t,d+1} = 1$ and $h_{t,d+1} = *$ if $s_{t,d+1} = 0$. That is, $*$ is the symbol for not observing a feature or the label. The next lemma, which is the key lemma of the proof, shows that the expected value of $N_i$ does not change too much if we change the opponent strategy.

**Lemma A.3.2.** (Bartók 2012, Lemma 26) *There exists a constant $C_1$ such that for any $i, j \in \{1, \dots, d\}$,*

$$\mathbb{E}_i[N_i] - \mathbb{E}_j[N_i] \le C_1 T \varepsilon \sqrt{d \mathbb{E}_j[N_L]}\,.$$

Now we are equipped to lower bound the expected regret. Let

$$j = \operatorname{argmin}_{k \in \{1,\dots,d\}} \mathbb{E}_k[N_L].$$

By Lemma A.3.2,

$$\mathbb{E}_i[R_T] \ge (T - \mathbb{E}_i[N_i]) \frac{d\varepsilon}{2} + c_{d+1} \mathbb{E}_i[N_L]$$

$$\ge \left( T - \mathbb{E}_j[N_i] - C_1 T \varepsilon \sqrt{d \mathbb{E}_j[N_L]} \right) \frac{d\varepsilon}{2} + c_{d+1} \mathbb{E}_j[N_L]$$

Denoting $\sqrt{\mathbb{E}_j[N_L]}$ by $\nu$ we have

$$\frac{1}{d} \sum_{i=1}^{d} \mathbb{E}_i[R_T] \ge \left( T - \frac{1}{d} \sum_{i=1}^{d} \mathbb{E}_j[N_i] - C_1 T \varepsilon \sqrt{d} \nu \right) \frac{d\varepsilon}{2} + c_{d+1} \nu^2$$

$$\ge \left( T - \frac{T}{d} - C_1 T \varepsilon \sqrt{d} \nu \right) \frac{d\varepsilon}{2} + c_{d+1} \nu^2$$

What is left is to optimize this bound in terms of $\nu$ and $\varepsilon$. Since $\nu$ is the property of the algorithm, we have to minimize the expression in $\nu$, with $\varepsilon$ as a parameter. After simple algebra we get

$$\nu_{opt} = \frac{C_1 T \varepsilon^2 d^{3/2}}{4 c_{d+1}}\,.$$

Substituting it back results in

$$\frac{1}{d} \sum_{i=1}^{d} \mathbb{E}_i[R_T] \ge (d-1) \frac{T\varepsilon}{2} - \frac{C_1^2 T^2 \varepsilon^4 d^3}{16 c_{d+1}}$$

Now we set

$$\varepsilon = \left( \frac{2}{C_1^2} \right)^{1/3} (c_{d+1})^{1/3} d^{-2/3} T^{-1/3}$$

to get

$$\mathbb{E}[R_T] \ge C_3 (c_{d+1})^{1/3} d^{1/3} T^{2/3}$$

whenever $d \ge 2$. $\qquad\square$