[Reviews · NeurIPS 2013]

Submitted by Assigned_Reviewer_3

Online Learning with Costly Features and Labels

Summary:
The paper discusses a version of sequential prediction where there is a cost associated to obtaining features and labels. Two cases are considered. First, the case where labels are given but features are bought. Here the regret bound is of the form sqrt{2^d T}. The time dependence is as desired, and a lower bound shows that the exponential dependence in the dimension of the features space cannot be reduced in general. In the second case the labels are also assumed costly. Here the regret dependence on the time horizon explodes to the T^{2/3} regime.

Discussion:
This paper is well structured and easy to follow. The considered problem of "feature efficient prediction" is a natural variant of "label efficient prediction". The paper provides a detailed analysis with both upper and lower bounds, showing that there is a complexity transition when labels become unobserved. The pros and cons (computational complexity) are discussed in balance.



Medium things:
You choose your curly F' to be a subset of all mappings, not of the original set curly F. If this is an oversight I would prefer you change it. If it does make a difference I think this needs a bit more discussion.

In section 2.2 it would be good to state and compare with the full information regret.

In Theorem 2.4, line 365, something goes awry with the notation. I think things would be fixed by the substitution ylim = wlim*xlim, X_1=xlim, W_\infty=wlim.

In section 2.2.1, line 377, I would like to see a little more explanation about the problems with "small sampling probabilities". What is small? Why is the matrix not positive semi-definite in that case, and why is this a problem?

A.2.2: I would like to see a little more guidance as to how a bandit lower bound using worst-case loss functions gives a result for quadratic loss functions.



Small things:

line 090: "the loss function is often compares" omit "is".

line 109: The argument why querying labels only already provides the regret lower bound is currently written down as if the reader should be able to see this too. But that is not the case for me. I guess an "...as we show..." would help.

line 186: "explore less exploiting" -> "explore less, exploiting"

line 208: There is a stray $T > 0$ in this line?

line 214: This line would make more sense without "sens"es

line 223: The exponent d should be placed on the set in line 224 instead.
line 223: The set should have a vertical bar where there is a comma.

lines 228 and 232, Theorem 2.2: Why does the lower order term not scale with any of the range parameters (CELP, C1, lmax)? Would this imply it can be made arbitrarily small (and hence removed) by pre-scaling the problem?

line 224: You only use origin-centred balls. You even forgot "x" in the definition. I guess the centre argument can safely be dropped.

line 255: "L\in" -> "\Lin"

line 296: "it suffices if we" -> "it suffices to"

line 247: "adding feature cost term" -> "adding a/the feature cost term"

line 303: not all (feature support of) "s"-ses are upright here and later. Please make them consistent.

line 362: curly F_lin is not defined. This used to be Lin?

line 404: "case we need approximately" drop "we need"

line 409: "discretize" -> "discretized"

line 411: You now omit the 1+ from the logarithm, but I fail to see where it went.

Summary: This paper considers problems where feature values are costly, and settles most of the questions.

Submitted by Assigned_Reviewer_4

This paper studies a new problem named the online probing. In each round of the learning process, the learner can purchase the values of a subset of feature values. After the learner uses this information to come up with a prediction for the given round, he has the option of paying for seeing the loss that he is evaluated against. The authors provide detailed prove on a number of theories in the new problem.

On the up side, this paper addresses a new problem for machine learning. Also, the paper is reasonably well written.

However, my main concern is that any new problem should have certain practical applications, even on synthetic data. However, this paper has no mention of the application, and no experiments at all. This is a major problem and it is hard to convince me the value of the proved theories.
Summary: I tend to vote against this paper due to luck of any experiments.

Submitted by Meta_Reviewer_7

Online Learning with Costly Features and Labels
------------------------------------------
Scope: The paper considers the problem of online learning where the learner pays for observing certain features as well as the label/loss. The main results include a T^{1/2} regret algorithm for the feature-cost setting with free labels, and T^{2/3} lower and upper bounds for the setting in which the labels incur a cost as well.

The setting is of definite interest. The algorithms and lower bounds are pretty simple technically. The root(T) upper bound follows from Mannor-Shamir, and the T^{2/3} bounds are not surprising to the expert online-learner. Still, this reviewer is of the opinion that simplicity is not necessarily a downside, given that the model and topic are sufficiently novel.

Several similar results that deal with attribute effiency have appeared in the learning literature of late, most are surveyed in the paper. However, one particular result which seems to subsume the poly(d) root(T) upper bound for the square loss at least partially is this: Elad Hazan, Tomer Koren: Linear Regression with Limited Observation. ICML 2012 (the algorithm itself appears similar)
Summary: Nice extension of attribute-efficient prediction. Simple but elegant algorithms and lower bounds that strongly build upon previous work.
Author Feedback

Author rebuttal: We thank the reviewers for their insightful comments. Below, we address the main concerns raised by the reviewers.

Reviewer_3:

Thanks for the very careful reading of the paper - we really do appreciate it. Some small clarifying remarks (when we do not reply to a comment, we agree, and we will implement the requested change):

Curly F’ does not need to be a subset of curly F, but in practice it is often chosen to be a subset because it is natural to do so, or sometimes, it is a requirement. However, in our framework the learner chooses a prediction and not a predictor from curly F, so this is not a requirement.

Theorem 2.4, line 365: indeed, the suggestion would work. Even better is to define \ylim to be a bound on |y_t|.

Section 2.2.1, line 377: If the loss function estimate was always convex, we could use methods designed for the problem of online linear optimization with bandit feedback, which would result in algorithms whose regret bound would be of optimal \sqrt{T} rate, and both the regret and the computational complexity would scale polynomially in the dimension d. This would be a huge improvement over our current algorithms. However, the following simple example shows that \tilde{X} is not always positive semi-definite -- equivalently, our loss function estimate is not necessarily convex: Assume we only have two features, both with value 1 (unknown to the algorithm), and the features are sampled independently with probability q<1. Then, with probability 1/q^2, we use both features, and \tilde{X} becomes [1/q, 1/q^2; 1/q^2, 1/q], with determinant 1/q^2-1/q^4, which is negative for all q<1 (in this sense the expression "sufficiently small sampling probability" might indeed have been misleading).
As an alternative to our algorithms, we could use an MCMC approximation of the continuous exponentially weighted average prediction method. If \tilde{X} was positive semi-definite, this would require sampling from truncated Gaussian distributions. When \tilde{X} is not positive semi-definite, or, more precisely, a submatrix of \tilde{X} is negative definite, sampling in the corresponding directions becomes complicated, as most of the mass is concentrated on the boundary of the constraint set. In our current research we work on designing efficient sampling algorithms for such distributions.


lines 228 and 232, Theorem 2.2: The lower order terms comes from the approximation error. The parameter alpha hides some of the constants.

Reviewer_4:

The proposed problem is motivated by several applications as discussed in lines 61-80 in the submitted paper. Furthermore, according to the call for papers, NIPS welcomes learning theory papers, and we do not think that it is a requirement for a learning theory paper to have experimental results.



Reviewer_6:

We did not mention [Hazan and Koren, ICML 2012] as that model is quite different. The revised version will include a discussion of the differences, especially that they consider a statistical setting, which differs significantly from our focus on an adversarial setting. Moreover, they consider features are costly only during the training phase and they do not consider feature costs when evaluating the performance of a predictor.